# MicroRNA 148a Suppresses Tuberculous Fibrosis by Targeting NOX4 and POLDIP2

**DOI:** 10.3390/ijms23062999

**Published:** 2022-03-10

**Authors:** Seong Ji Woo, Youngmi Kim, Harry Jung, Jae Jun Lee, Ji Young Hong

**Affiliations:** 1Institute of New Frontier Research Team, Hallym University College of Medicine, Chuncheon 24253, Korea; seong-jikr@nate.com (S.J.W.); kym8389@hanmail.net (Y.K.); harry880219@gmail.com (H.J.); 2Division of Pulmonary, Allergy and Critical Care Medicine, Department of Internal Medicine, Chuncheon Sacred Heart Hospital, Hallym University Medical Center, Chuncheon 24253, Korea

**Keywords:** tuberculosis, fibrosis, NADPH oxidase 4, microRNA

## Abstract

Extracellular matrix production by pleural mesothelial cells in response to *Mycobacterium tuberculosis* contributes to tuberculous fibrosis. NOX4 is involved in the pathogenesis of tuberculous fibrosis. In this study, we evaluated whether *NOX4* gene-targeting microRNAs showed protective effects in tuberculosis fibrosis. TargetScan prediction software was used to identify candidate microRNAs that bind the 3′ UTRs of *NOX4*, and microRNA-148a (miR-148a) was selected as the best miRNA candidate. A repressed and forced expression assay in Met5A cells was performed to investigate the causal relationship between miR-148a and NOX4. The role of miR-148a in tuberculous pleural fibrosis was studied using a murine model of *Mycobacterium bovis* bacillus Calmette–Guérin (BCG) pleural infection. Heat-killed *M. tuberculosis* (HKMT) induces NOX4 and POLDIP2 expression. We demonstrated the inhibitory effect of miR-148a on NOX4 and POLDIP2 expression. The increased expression of miR-148a suppressed HKMT-induced collagen-1A synthesis in PMC cells. In the BCG pleurisy model, miR-148a significantly reduced fibrogenesis and epithelial mesenchymal transition. High levels of miR-148a in tuberculous pleural effusion can be interpreted as a self-limiting homeostatic response. Our data indicate that miR-148a may protect against tuberculous pleural fibrosis by regulating NOX4 and POLDIP2.

## 1. Introduction

Tuberculosis (TB) is a global public health problem responsible for 9.9 million incident cases [1]. TB mortality has reduced, and 85% (66 million cases) of the treated patients survive [1]. However, approximately half of the TB survivors complain of chronic pulmonary dysfunction and reduced quality of life despite microbiological cures [2,3,4]. This is associated with excessive deposition of the extracellular matrix (ECM), stiffness, and parenchymal scarring in tuberculous fibrosis [5,6]. Residual distortion of the lung architecture resulting in pulmonary dysfunction depends on the degree to which the ECM is degraded [7]. Investigating the underlying immunological mechanism involving tuberculous fibrosis could help identify therapeutic targets.

Nicotinamide adenine dinucleotide phosphate (NADPH) oxidase 4 (NOX4) is the main source of reactive oxygen species (ROS) and is upregulated in several lung diseases, including idiopathic pulmonary fibrosis, bleomycin-induced lung injury, and lung cancer [8,9,10,11,12]. We recently reported that NOX4 signaling regulates ERK–ROS signaling and EMT pathways, thereby contributing to tuberculous fibrosis [13]. Antagonizing NOX4 expression restores impaired protective autophagy in tuberculous pleurisy.

MicroRNAs (miRNAs) are short, single stranded, noncoding RNA molecules composed of approximately 19–24 nucleotides [14]. miRNAs downregulate target gene expression mainly by binding to the consensus sequence of the target mRNA and inactivating the target gene using transcription factor-aided co-action [14,15,16]. Therefore, miRNAs have great potential as a therapeutic alternative in cancer, neurodegenerative disorders, cardiovascular diseases, and infectious diseases [17]. While some miRNAs are involved in the pathogenesis of tuberculosis, few studies have explored the role of miRNAs in tuberculous fibrosis [18].

In the present study, we sought to identify potential miRNA that regulate NOX4 in tuberculous fibrosis and identify new therapeutic targets. Among miRNAs predicted to bind to NOX4 by bioinformatics analysis, miR-148a was evaluated. To elucidate the miR-148a–related mechanism in tuberculous fibrosis, a mesothelial cell experiment and BCG pleurisy mouse model were performed. The expression of miR-148a in tuberculous pleural effusion, a clinical specimen corresponding to a chronic condition after tuberculous infection, was compared with transudate.

## 2. Results

### 2.1. NOX4 and POLDIP2 Expression in Mesothelial Cells Is Upregulated after Treatment with HKMT

Our previous study showed that HKMT treatment enhanced the expression of NOX4, POLDIP2, and collagen-1A in mesothelial cells (Figure 1A). HKMT treatment increased the expression of NOX4 and POLDIP2 in an interdependent manner. The downregulation of NOX4 decreased the expression of POLDIP2 and vice versa (Figure 1B,C). Immunoprecipitation verified the direct interaction between NOX4 and POLDIP2 (Figure 1D).

### 2.2. miR-148a Is Underexpressed during the Initial Stage of Exposure to Tuberculosis but Increases Later

The TargetScan prediction software identified several miRNAs that are broadly conserved among vertebrates and putatively bind to sites in the untranslated regions (3′ UTRs) of NOX4 mRNA (Appendix A). We evaluated the expression of candidate miRNAs in mouse pleural mesothelial cells (mPMCs) obtained from Wild type (WT) or NOX4 KO mice, with and without the 3 h HKMT treatment (Figure 2A). Of the candidate miRNAs, miR-9, miR-148a and miR-196 were higher in the mPMCs of NOX4 KO mice than in WT mice. After 3 h HKMT treatment, the expression of miR-9 and miR-148a was downregulated in both WT mPMC and Met5A cells (Figure 2B, C). Knocking out NOX4 or SiRNA NOX4 treatment did not reverse the decrease in miR-9 and miR-148 induced by 3 h HKMT treatment. 

We evaluated the changes in miRNAs over time after HKMT treatment in human Met5A cells and mPMCs (Figure 3A–D). miR-148a showed a clear trend over time. miR-148a expression decreased initially after HKMT treatment but tended to increase after long-term exposure. NOX4 and miR-148a showed differences in expression over time. In particular, in the mPMC experiment, NOX4 increased at 24 h and then decreased at 72 h, whereas miR-148a decreased at 24 h and then showed a marked increase at 72 h. This change of miR-148a encouraged us to focus on miR-148a as a potential candidate for the regulation of NOX4 in tuberculous fibrosis. The binding site of miR-148a was computationally predicted in the 3′ UTRs of NOX4 mRNA (Appendix A).

TB-PE indicates a chronic status after TB exposure. Consistent with Met5A with long term HKMT exposure, the expression of circulating miR-148 was higher in TB-PE than in the transudate (Figure 3E).

### 2.3. miR-148a Regulates Expression of NOX4/POLDIP2 in Met5A Cells

To validate the correlation between NOX4 and miR-148a in tuberculous fibrosis, we investigated whether the expression level of miR-148a affects the expression level of NOX4 using a synthetic mimic and inhibitor of miR-148a. Met5A cells were transfected with miR-148a mimic or miR-148a inhibitor and treated with HKMT for 3 h. Figure 4b shows that increasing miR-148a levels remarkably decreased HKMT-induced transcription of NOX4 and collagen-1A. Conversely, miR-148a knockdown using miR148a-inhibitor significantly increased the levels of NOX4, POLDIP2, and collagen-1A (Figure 4C).

HKMT treatment upregulated NF-κB, a transcription factor of NOX4, and miR-148a mimics downregulated HKMT-induced NF-κB (Figure 4D). The ChIP assay demonstrated that HKMT increased the binding of NOX4/POLDIP2 to the miR-148a promoter-1 sequence (Figure 4E). These results suggest that NOX4 may mediate miR-148a downregulation in tuberculosis fibrosis by directly regulating gene transcription. Taken together, these results suggest that there is a mutually inhibitory relationship between NOX4 and miR-148a.

### 2.4. miR-148a Prevents Experimental Tuberculous Fibrosis

An increase in miR-148a was observed 2 days after miRNA tail vein injection, compared to control mice (Appendix A). To determine the potential role of miR-148a in the development of tuberculous fibrosis, miR-148a mimics or control mimics (20 nmol/kg) was administered once every two days for 2 weeks after BCG injection (Figure 5A). After 2 weeks, the expression of miR-148a in lung tissue was higher in the BCG + control (Ctrl) group than in the PBS + Ctrl control group and was highest in the BCG + miR-148a mimic group among the four groups (Figure 5B). The administration of miR-148a mimics significantly attenuated BCG-induced expression of NOX4, POLDIP2, and Snail proteins (Figure 5C). To examine the effects of miR-148a mimic post-treatment in BCG-induced pleurisy, cell recruitment and cytokine accumulation were determined. On day 15 after infection, there was a significant increase in the total number of pleural cells (PBS + Ctrl: 21.3 × 10^4^ ± 12.74 × 10^4^/mL, BCG + Ctrl: 786.7 × 10^4^ ± 369.5 × 10^4^/mL, *p* = 0.023) (Figure 5D). The total count in mice in the BCG + miR-148a mimic group was lower than that in the BCG + Ctrl group (BCG + miR-148a mimic: 232.8 × 10^4^ ± 313.0 × 10^4^/mL, *p* = 0.084). The microscopic analysis of pleural cells showed that the number of multinucleated giant macrophages in mice was higher in the BCG + Ctrl group than in the PBS + Ctrl and BCG + miR-148a mimic groups (Figure 5E). Similarly, the concentration of cytokines (IL-6, TNF-α and IFN-ɤ) measured in the serum was higher in the BCG + Ctrl group than PBS + Ctrl group and miR-148a mimic post treatment reduced the expression of cytokines induced by BCG (Figure 5F). These results suggest that miR-148a regulates NOX4 expression levels and leads to substantial control of pleural inflammation as well as fibrosis in BCG pleurisy.

Hematoxylin and Eosin (H&E) staining showed an improvement in BCG-induced fibrosis in mice with increased miR-148a expression. Fifteen days after BCG infection, all mice in the BCG + Ctrl group showed increased pleural hyperplasia and pulmonary cell infiltration compared to the PBS + Ctrl group. Mice in the BCG + miR-148a mimic group presented with reduced submesothelia, granulomas, and pleural hyperplasia compared to mice in the BCG + Ctrl group (Figure 6A). Immunostaining of the mouse lung tissue for NOX4 showed remarkably strong expression around infiltrative areas with thickened septae in the BCG + Ctrl group, contrary to the expression in the PBS + Ctrl group, which showed weak detectable expression in epithelial bronchial cells. Mice in the BCG + miR-148a mimic group showed reduced NOX4 expression in lung tissue (Figure 6B). Collagen deposits were also observed in the pleural sections of mice in the BCG + Ctrl group, but mice in the BCG + miR-148a mimic group had fewer collagen deposits compared to mice in the BCG + Ctrl group (Figure 6C). The in vivo suppressive effect of miR-148a on NOX4 was verified using immunofluorescence staining (Figure 6D). We found increased levels of NOX4, which were colocalized with mesothelin, a marker of mesothelial cells in the BCG + Ctrl group than in the PBS + Ctrl group. The miR-148a mimic treatment blocked BCG induced mesothelin and NOX4 expression.

## 3. Discussion

In this study, miR-148a was identified as a novel miRNA involved in tuberculous fibrosis.

In accordance with previous studies [13,19], our data demonstrate that NOX4 signaling contributes to tuberculous fibrosis and interacts with POLDIP2. Using bioinformatics analysis, we identified NOX4 as a potential target gene of miR-148a. We report that HKMT downregulated the expression of miR-148a initially and upregulated the expression of miR-148a after exposure for a long time. The same results were confirmed in clinical specimens. miR-148a showed a tendency to be higher in the tuberculous pleural effusion, compared with transudate. Functionally, miR-148a mimics reversed the HKMT induced upregulation of NOX4, POLDIP2, and collagen-1A mRNA. Likewise, in an in vivo model of BCG pleurisy, miR-148a mimics decreased the production of proinflammatory cytokines and deposition of ECM by inhibiting NOX4. This protective role was confirmed in mouse models with upregulated expression of miR-148a. We speculate that miR-148a could effectively attenuate fibrosis after tuberculous infection. In addition, this result indicates that miR-148a upregulation in tuberculous pleural effusion is a protective mechanism or homeostasis response.

Recent studies have raised the possibility that circulating miRNAs in body fluids may contribute to the early diagnosis and monitoring of treatment response in various diseases [20,21]. In addition, miRNAs are more stable and have higher sensitivity and specificity compared to protein biomarkers [22]. Some miRNAs have been proposed to be associated with ECM repair in chronic fibrotic lung diseases or to the clinical outcomes of TB [18,23]. Little is known about the involvement of miR-148a in fibrotic lung disease. There have been reports that miR-148a suppresses lung cancer and liver fibrosis [24,25]. miR-148a suppressed the migration and invasion of non-small cell lung cancer cells by targeting Wnt1 signaling [26]. The overexpression of miR-148a downregulates ERBB3, which is required for the activation of hepatic stellate cells and liver fibrosis [25]. Miotto et al. demonstrated that miR-148a was present in serum miRNA signatures could be used to differentiate active TB and healthy subjects’ patients [27]. Whether pleural effusion or plasma miR-148a can be a useful biomarker for the identification of tuberculous fibrosis requires validation studies using multiple clinical samples.

We did not demonstrate direct binding of miR-148a to NOX4, but our data suggest an indirect regulation of miR-148a and NOX4. Several transcription factors, such as NFκB, SMAD2/3, E2F, HIF1α, and Nrf2 regulate NOX4 promoter activity [28]. NF-κB stimulates NADPH oxidases in human phagocytes and vascular smooth muscle cells [29,30]. Lu et al. reported that hypoxia increased the binding of the NF-kB subunit P65 to the NOX4 promoter, and PPARγ attenuated this binding as a negative regulator [31]. In our study, miR-148a mimics reduced the expression of NF-κB, a transcription factor of NOX4, leading to the downregulation of NOX4 levels. The detailed mechanisms of the interaction between miR-148a and NF-κB remain to be elucidated.

In addition, the ChIP assay demonstrated that NOX4/POLDIP2 is bound to a specific site on the promoter of the miR-148a gene and that the binding was enhanced by HKMT treatment. Taken together, our data support the existence of auto-regulatory feedback between NOX4 and miR-148a as a microRNA/target protein network. TB fibrosis activates miR-148a to repress NOX4, which conversely inhibits miR-148a, providing a self-limiting protective mechanism (Figure 7).

We previously reported an association between TB fibrosis and tumorigenic potential [32]. Interestingly, miR-148a has been reported to inhibit lung cancer [24]. Considering the antifibrosis function of miR-148a shown in our data, this may reflect the crosstalk mechanism between fibrosis and cancer. Further studies are needed to verify whether tuberculous fibrosis related miRNAs mediate cancer cell proliferation. In addition, validation of this finding should be performed to examine clinical utility in a larger cohort.

The limitation of this study is that it is difficult to extrapolate the results to human tuberculosis because a mouse model of BCG-induced pleurisy is not identical to human tuberculosis. However, HKMT was used to study tuberculosis in vitro experiments.

In conclusion, miR-148a can attenuate tuberculous fibrosis by suppressing NOX4 by downregulating the transcription factor NF-κB in the mesothelial cell experiment. Notably, NOX4 downregulates miR-148a, resulting in a negative feedback loop. Likewise, in the mouse model, we reconfirmed the contribution of miR-148a in the attenuation of pleural fibrosis associated with BCG-induced pleurisy. The clinical potential of miR-148a should be further evaluated in future studies.

## 4. Materials and Methods

### 4.1. Cell Lines and Animals

The human mesothelial cell line Met5A was purchased from the American Type Culture Collection (ATCC, Manassas, VA, USA). The wild type NOX4 (NOX4-WT) mice and knockout NOX4 (NOX4-KO) C57BL/6 mice were obtained from Prof. Park (Yonsei University, Seoul, Korea). All animal experiments were approved by the Institutional Animal Care and Use Committee (IACUC) of Hallym University (NO; Hallym 2020-21).

### 4.2. Isolating Mouse Pleural Mesothelial Cells (mPMCs)

Primary mPMCs were obtained from the surface of the thoracic diaphragm, heart, and lungs of NOX4-WT or NOX-KO mice (3–4 weeks of age). Each tissue was separated and incubated in 0.25% trypsin-EDTA solution for 1 h after washing with Dulbecco’s phosphate buffered saline (DPBS). After centrifugation for 3 min at 1500 rpm, the cell pellets were suspended in Dulbecco’s modified Eagle medium (DMEM) (15% FBS, 1% penicillin/streptomycin) and cultured for one day in a 5% CO2 incubator. The next day, the cells attached to the dish were washed with DPBS and grown in DMEM (15% FBS, 1% penicillin/streptomycin) for 2–3 days. The non-attached cells were attached back to the plate and used 2–3 days later.

### 4.3. Treatment of Heat-Killed Mycobacterium Tuberculosis (HKMT)

HKMT (InvivoGen, San Diego, CA, USA) was used to study tuberculosis in Met5A and mPMCs. The cells at 60–70% confluence were treated with 10 ng/mL HKMT either with or without transfection for the indicated time periods. To suppress endogenous gene expression, siNOX4 (sense 5′-CUGUUGUGGACCCAAUUCA-3′ and antisense 5′ UGAAUUGGGUCCACAACAG 3′), siPoldip2 (sense 5′-CUCUUGUUCACUUUACCUU 3′ and antisense 5′ AAGGUAAAGUGAACAAGAG) were used in the experiments. Scrambled siRNA (sense: 5′-GGTCAAGACACTATTAACA-3′ and antisense: 50-GGATTCCTAGTGT ATTTCA-3′) was used as a control (SiCtrl). To determine the effect of miR-148a on in vitro tuberculosis, 200 pmol miR-148a mimic (5′-AAAGUUCUGAGACAC UCCGACU-3′) or miR-148a inhibitor (5′-UCAGUGCACUACAGAACUUUGU-3′), control mimic and control inhibitor were transfected into cells using the transfection reagent in jet PRIME transfection (PolyPlus Transfection, New York, NY, USA).

### 4.4. The Identification of Potential miRNAs

The TargetScan program was used to identify putative miRNAs predicted to regulate gene expression by directly binding to the 3′ untranslated regions (3′ UTRs) of NOX4 (Appendix A).

### 4.5. Immunoblot and Immunoprecipitation (IP) Analysis

Immunoblot analysis and immunoprecipitation were performed according to standard procedures as previously described [33,34]. Briefly, cells were lysed in RIPA buffer (50 mM Tris, pH 7.4, 150 mM NaCl, 0.5% sodium deoxycholate, 0.1% sodium dodecyl sulphate (SDS), and 1% Nonidet-P40). For immunoblotting, the protein lysates were mixed with 5X sample buffer, blended for 8 min, and 50 µg of protein was separated using 10% SDS polyacrylamide gel electrophoresis (SDS PAGE) and transferred to PVDF membranes (Thermo Scientific, Waltham, MA, USA). Membranes were incubated with primary antibodies against NOX4 (1:2000 dilution; Proteintech, Wuhan, China), POLDIP2, ZO-1 (1:1000 dilution; Abcam, Cambridge, UK), Snail (1:1000 dilution; Cell Signaling Technology, Danvers, MA, USA), followed by incubation with an HRP-conjugated secondary antibody (1:1000 dilution; Thermo Scientific). The membranes were stripped and reprobed with a primary antibody against β-actin (1:1000 dilution; Cell Signaling Technology).

For immunoprecipitation, cell lysates (500 µg) were immunoprecipitated with respective primary antibodies (2 µg) overnight at 4 °C; 40 µL of Protein A/G PLUS-Agarose (Santa Cruz Biotechnology, Santa Cruz, CA, USA) was added and the lysates were further incubated for 1 h at 4 °C. After washing the beads three times with lysis buffer, 2X sample buffer was added. The samples were boiled for 8 min and analyzed using 10% SDS-PAGE, followed by immunoblotting.

### 4.6. Quantitative Real Time Polymerase Chain Reaction (qRT-PCR)

Total RNA samples were extracted from Met5A and mPMCs using the easy-Blue reagent (iNtRON Biotechnology, Seoul, Korea). Strand cDNA was synthesized using the Maxime RT PreMix Kit (iNtRON Biotechnology). Total miRNA was extracted from frozen mouse lung tissue and Met5A or mPMCs using miRNeasy Mini Kit (Qiagen, Germantown, MD, USA), followed by reverse transcription using the miScript Transcription Kit (Qiagen). For quantitative RT-PCR (qRT-PCR), the SYBR Green PCR Kit (Qiagen) was used in Rotor-Gene Q (Qiagen). The primer sequences used for qRT-PCR are shown in Table 1 and Table 2.

### 4.7. Chromatin Immunoprecipitation Assay (ChIP Assay)

A ChIP assay was performed according to the manufacturer’s instructions (Cell Signaling Technology) on cultured Met5A cells. Briefly, the cells were crosslinked with 1% formaldehyde at 37 °C for 10 min. Chromatin extracts were isolated and further fragmented by sonication on ice to shear DNA. The nuclear extracts were immunoprecipitated with either anti-NOX4 antibody, anti-POLDIP2 antibody, or rabbit IgG at 4 °C overnight. Purified immunoprecipitated DNA/protein complexes were subjected to PCR. To identify the binding sites of proteins to the miR-148a promoter, three pairs of specific primers were synthesized to amplify DNA fragments of approximately 150–200 bp in the promoter region of miR-148. The primer sequences used for PCR were as follows: miR-148a promoter-1 sequences [5′-CAGCACGAGGAACTTGACCCA-3′ (sense) and 5′-GCAAGGC ACTGCACACACTAAC-3′(antisense)], miR-148a promoter-2 sequences [5′-TGGAGGTTT GGGTTGGTGAG-3′ (sense) and 5′-TACCAAGGGCTTCCCAGAGA-3′(antisense)], miR-148a promoter-3 sequences [5′-GCAGTGCCTTGCAGGAATTT-3′ (sense) and 5′-ATCTCCACAGCCCAAAAGCA-3′(antisense)]. IgG was used as a negative control. 

### 4.8. Enzyme-Linked Immunosorbent Assay (ELISA)

The expression of IL-6, TNF-α, and IFN-γ (Abbkine Wuhan, China) in mouse serum was measured using ELISA Kit.

### 4.9. Establishment of BCG-Induced Pleurisy and In Vivo Treatment of miRNA Mimics

Before miRNA 148a mimic treatment, 10^6^ CFUs BCG Pasteur in 100 µL saline into the intrapleural cavity was performed on day 0 to create a BCG-induced pleurisy animal model [13]. To determine the effect of miR-148a on in vivo tuberculous fibrosis, a control mimic or miR-148a mimic (20 nmol/kg) combined with 100 µL of in vivo-JetPEI (PolyPlus Transfection, New York, NY, USA) transfection reagent was injected intravenously once every two days from day 0 in a total of 14 days. On day 15, all mice were sacrificed, and lung tissues and blood serum were collected. Thoracic cavities were washed with 1 mL 2 mmol/L EDTA-PBS before sacrifice. The pleural effusion was centrifuged, and the cell pellet was reconstituted in 100 µL PBS. Counting of total cell numbers and cytospin with diff-Quick staining was performed as previously described [13].

### 4.10. Histology, Immunohistochemistry, and Immunofluorescence Staining

Paraffin sections of lung tissue were stained with hematoxylin and eosin (H&E) for lung inflammation. To evaluate pulmonary fibrosis, samples were stained with Masson’s trichrome stain kit (IMEB Inc., Chicago, IL, USA). After deparaffinization, lung tissues were immunostained with NOX4 antibody (Cusabio, Wuhan, China). For immunofluorescence staining, the sections were incubated with antibodies against NOX4 (1:100 dilution, Santa Cruz Biotechnology) and antibodies against mesothelin (1:100 dilution, Santa Cruz Biotechnology) overnight at 4 °C. After washing with PBS, the slides were incubated with Alexa Fluor 546-conjugated goat anti-mouse IgG1, Alexa Fluor 488-conjugated goat anti-rabbit IgG secondary antibody for 1 h in the dark and DAPI (Sigma-Aldrich, St Louis, MO, USA). All samples were examined under a fluorescence microscope (Olympus FV500; Olympus, Tokyo, Japan).

### 4.11. Patients and Sample Collection

The patient samples in this study were obtained according to the study protocol approved by the Institutional Review Board of Chuncheon Sacred Heart Hospital (IRB no.2012-27. All study participants provided written informed consent for inclusion in the study. The data did not include any information that could have led to patient identification. Tuberculous pleural effusion (TPE) was defined by two criteria: positive culture results from the pleural fluid sample or pleural biopsy with mycobacterial histological features (*n* = 3). Transudate pleural effusion was defined using Light’s criteria (*n* = 3) [35]. Circulating miRNA was extracted from pleural effusion (*n* = 6) using the miRNeasy Serum/Plasma Advanced Kit (Qiagen) according to the manufacturer’s instructions. Appendix A shows the characteristics of the study participants.

### 4.12. Statistical Analysis

Data are expressed as mean ± SEM. Statistical comparisons were made using two-way analysis of variance, followed by GraphPad Prism5. The significance of differences between groups was determined using Student’s unpaired *t*-test. Statistical significance was set at *p* < 0.05.

## Figures and Tables

**Figure 1 ijms-23-02999-f001:**
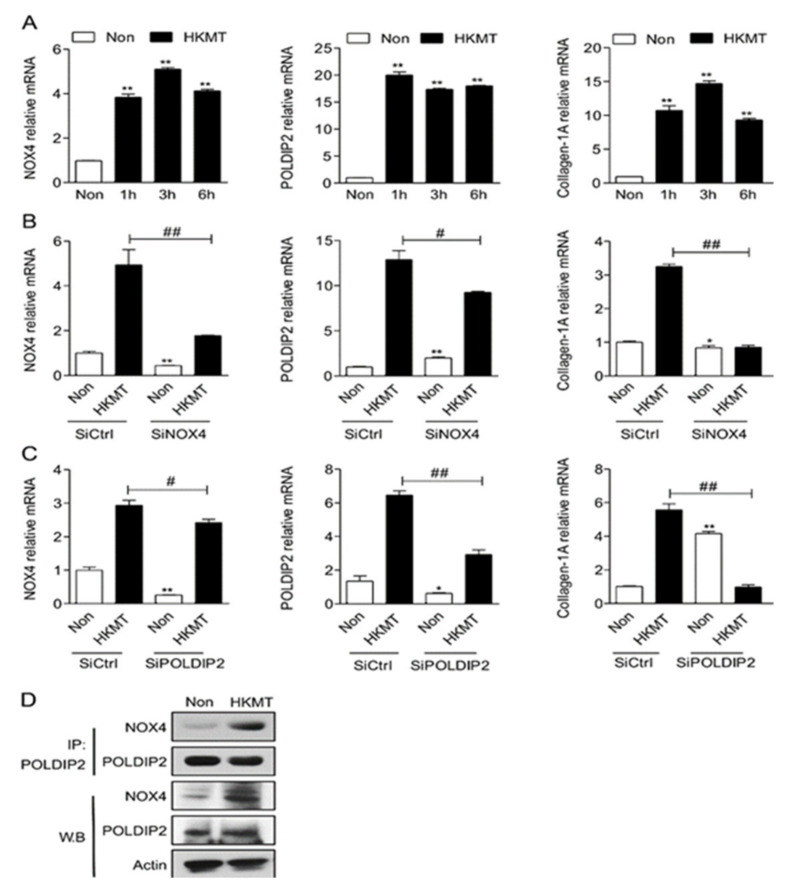
NOX4 and POLDIP2 mutually mediate HKMT-induced collagen-1A synthesis (**A**) NOX4, POLDIP2 and collagen-1A mRNA expression in the Met5A cells over time after HKMT treatment (**B**) NOX4, POLDIP2 and collagen-1A mRNA expression in the Met5A cells with HKMT treatment in the presence of SiCtrl and SiRNA targeting NOX4 (SiNOX4) (**C**) NOX4, POLDIP2 and collagen-1A mRNA expression in the Met5A cells with HKMT treatment in the presence of SiCtrl and SiRNA targeting POLDIP2 (SiPOLDIP2) (**D**) Interaction between NOX4 and POLDIP2 in Met5A cells after HKMT treatment confirmed by western blot and immunoprecipitation * significant difference (*p* < 0.05) compared to control group (SiCtrl), ** significant difference (*p* < 0.01) compared to (SiCtrl), # significant difference (*p* < 0.05) compared to HKMT group (SiCtrl + HKMT), ## significant difference (*p* < 0.01) compared to HKMT group (SiCtrl + HKMT).

**Figure 2 ijms-23-02999-f002:**
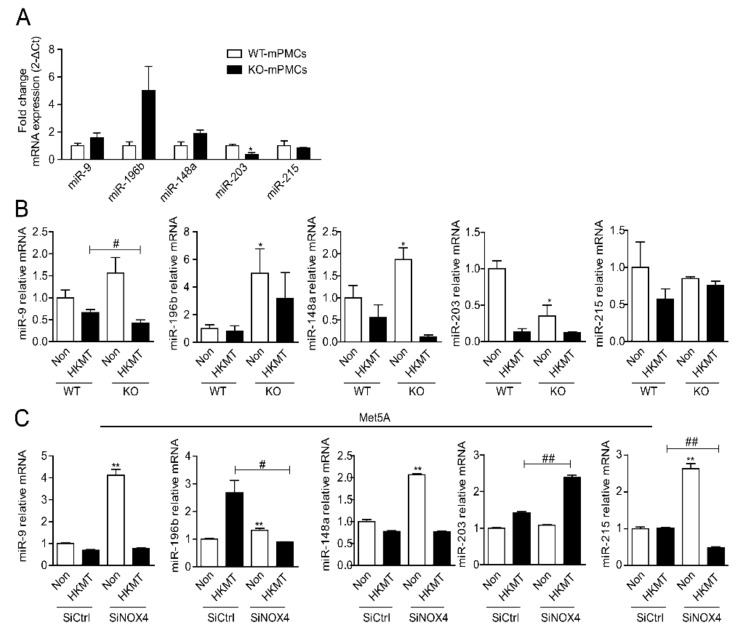
miRNA expression in mesothelial cells with HKMT treatment. (**A**) miRNA expression in mPMCs of WT mouse and NOX4 KO mouse (*n* = 3) (**B**) miRNA expression in mPMCs of WT mouse and NOX4 KO mouse following stimulation with and without HKMT. * significant difference (*p* < 0.05) compared to control group (WT mPMC Non), # significant difference (*p* < 0.05) compared to WT mPMC HKMT group (**C**) miRNA expression in human Met5A cells transfected with SiCtrl or SiNOX4 following stimulation with and without HKMT. ** significant difference (*p* < 0.01) compared to (SiCtrl), # significant difference (*p* < 0.05) compared to HKMT group (SiCtrl + HKMT), ## significant difference (*p* < 0.01) compared to HKMT group (SiCtrl + HKMT).

**Figure 3 ijms-23-02999-f003:**
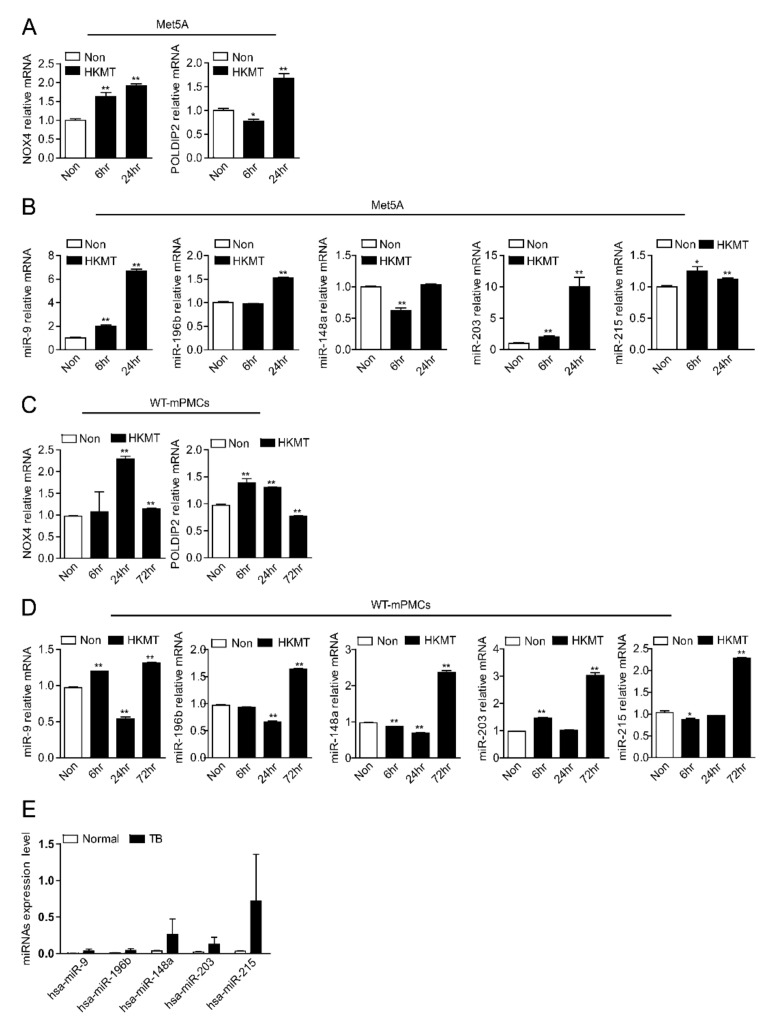
miR-148a decreased in the early stage of HKMT stimulation but increased as the exposure time increased. (**A**) NOX4, POLDIP2 mRNA expression in the Met5A cells pre-incubated with HKMT (10 ng/mL) for indicated times (*n* = 3). (**B**) qRT-PCR analysis of miRNA expression in Met5A cells pre-incubated with HKMT (10 ng/mL) for indicated times (*n* = 3). Results are presented as the mean ± SEM (**C**) NOX4, POLDIP2 mRNA expression in the WT-mPMCs pre-incubated with HKMT (10 ng/mL) for indicated times (*n* = 3). (**D**) qRT-PCR analysis of miRNA expression in WT-mPMCs pre-incubated with HKMT (10 ng/mL) for indicated times (*n* = 3). Results are presented as the mean ± SEM (**E**) The expression of circulating miRNA levels in tuberculous pleural effusion and transudate. * significant difference (*p* < 0.05) compared to control group (Non), ** significant difference (*p* < 0.01) compared to control group (Non).

**Figure 4 ijms-23-02999-f004:**
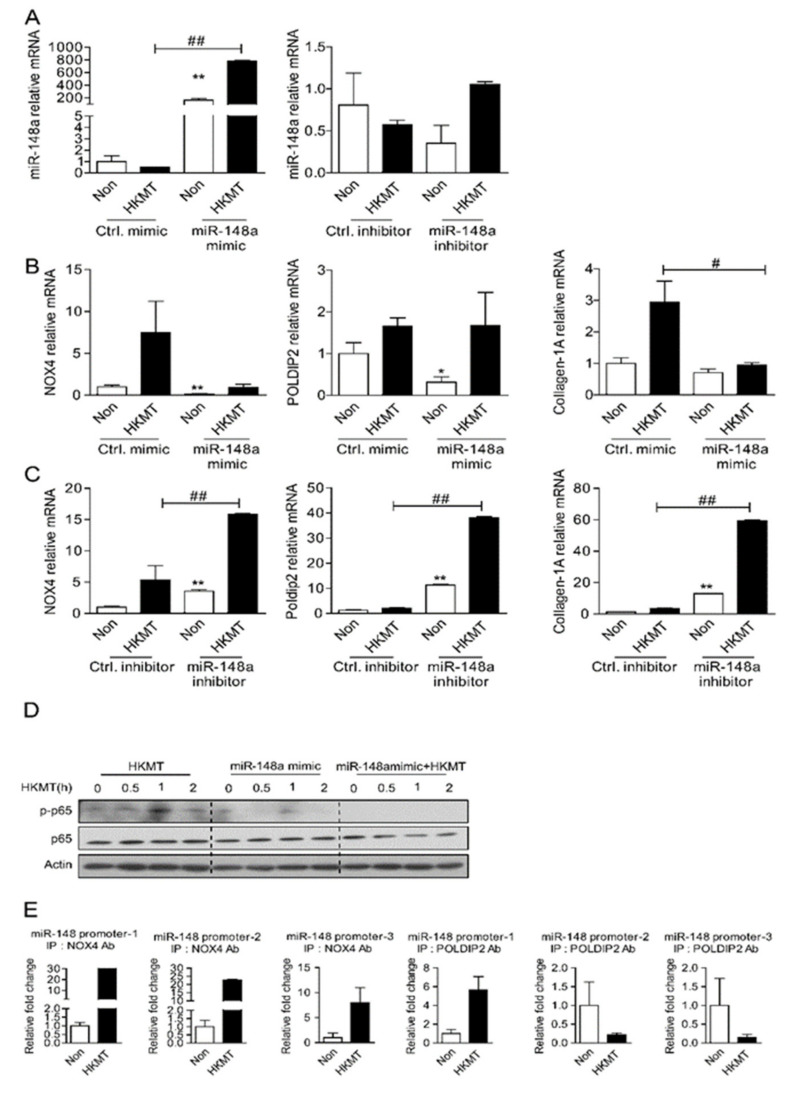
miR-148a can reduce tuberculous induced collagen-1A expression by regulating NOX4. (**A**) Levels of miR-148a in Met5A transfected with miR-148a mimics and miR-148 inhibitors in the presence or absence of HKMT treatment (**B**) NOX4 mRNA, POLDIP2 mRNA and collagen-1A mRNA in Met5A transfected with Ctrl mimics and miR-148a mimic in the presence or absence of HKMT treatment (**C**) NOX4 mRNA, POLDIP2 mRNA and collagen-1A mRNA in Met5A transfected with Ctrl inhibitor and miR-148 inhibitor in the presence or absence of HKMT treatment (**D**) NF-kB expression in Met5A transfected with miR-148a mimics in the presence or absence of HKMT treatment (**E**) ChiP assay of NF-kB binding in miR-148a promoter. Met5A cells were treated with or without HKMT for 3 h to collect the chromatin for immunoprecipitation with a specific anti-NOX4 antibody and a specific anti-POLDIP2 antibody. Quantitative data are expressed as mean ± SEM. * significant difference (*p* < 0.05) compared to control group (Ctrl mimic or Ctrl inhibitor), ** significant difference (*p* < 0.01) compared to control group (Ctrl mimic or Ctrl inhibitor), # significant difference (*p* < 0.05) compared to HKMT group (Ctrl mimic or Ctrl inhibitor + HKMT), ## significant difference (*p* < 0.01) compared to HKMT group (Ctrl mimic or Ctrl inhibitor + HKMT).

**Figure 5 ijms-23-02999-f005:**
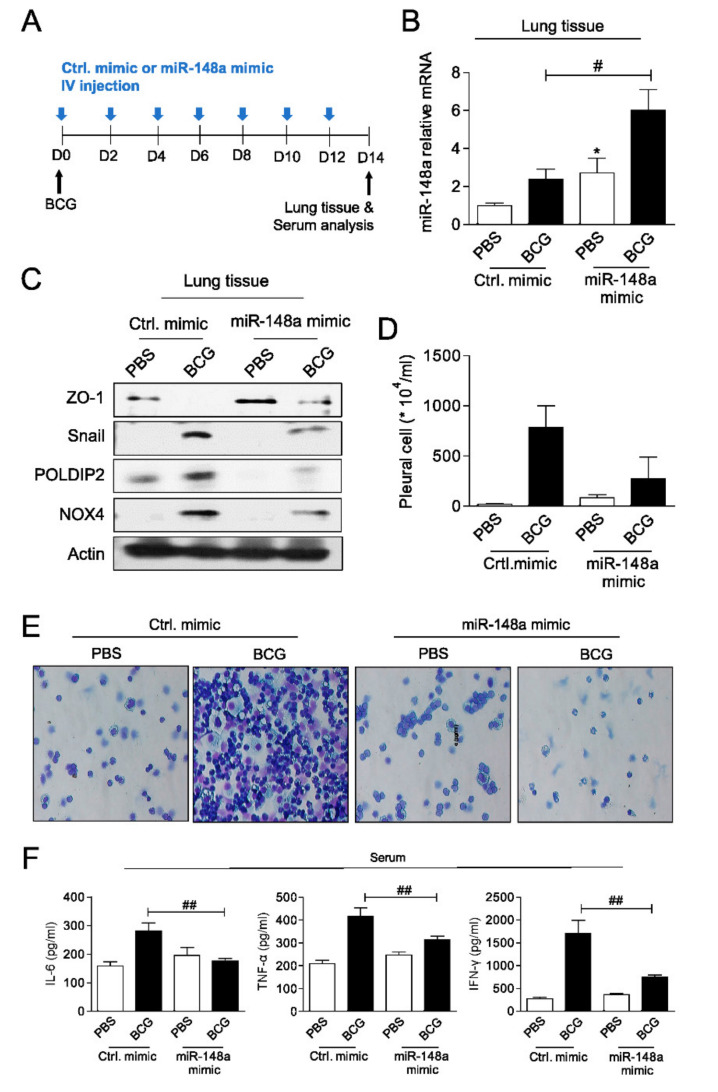
miR-148a prevents pulmonary fibrosis in mice (**A**) Schematic of experimental design (**B**) miR-148a expression (**C**) Immunoblots showing decreases of epithelial mesenchymal transition (EMT) markers, NOX4 and POLDIP2 in lung tissues of BCG treated mice with miR-148a mimics posttreatment (**D**) Total number of pleural cells in four groups (**E**) Photomicrographs from cytospin stained with Giemsa (**F**) Cytokine levels (IL-6, TNF-α and IFN-γ) assessed in the serum. * significant difference (*p* < 0.05) compared to control group (PBS + Ctrl. mimic), # significant difference (*p* < 0.05) compared to BCG + Ctrl. mimic group, ## significant difference (*p* < 0.01) compared to BCG + Ctrl. mimic group.

**Figure 6 ijms-23-02999-f006:**
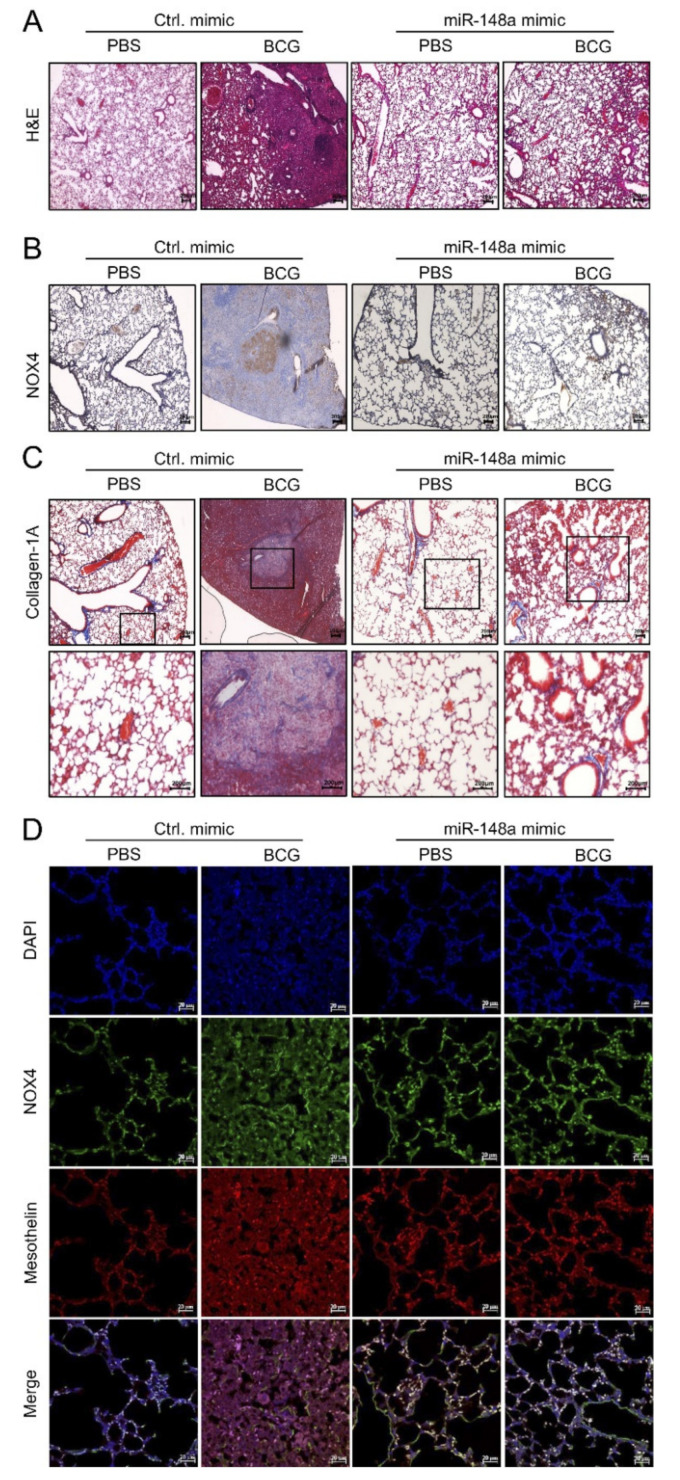
miR-148a posttreatment regulates NOX4 expression and inhibits collagen-1A infiltrates in mouse lungs. (**A**) H&E staining, ×100 (**B**) Immunostaining of NOX4, ×100 (**C**) Masson Trichrome staining in mouse lung. ×100, ×200 (**D**) Immunofluorescence staining of NOX4 and mesothelin (markers for mesothelial cells). ×400 Light yellow areas represent colocalization of NOX4 (green) and mesothelin (red). DAPI,4′,6-Diamidino-2-phenylindole dihydrochloride.

**Figure 7 ijms-23-02999-f007:**
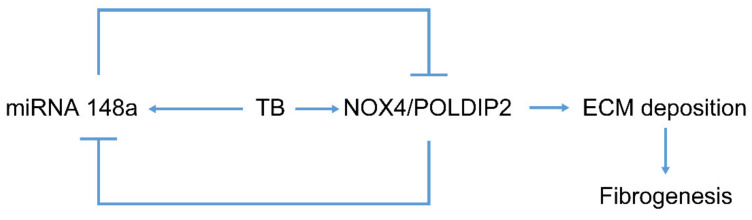
Regulation of tuberculous fibrosis by miR-148a. Chronic HKMT exposure induces miR-148a expression. Initially, TB downregulates miRNA 148a, but finally upregulates miR-148a. Overexpression of miR-148a in mesothelial cells blocked NOX4 and POLDIP2 expression, thus preventing ECM deposition and lung fibrogenesis. NOX4 may also inhibit the expression of miR-148a, providing a negative feedback mechanism.

**Table 1 ijms-23-02999-t001:** Human and mouse sequences and accession numbers for primers (forward, FOR; reverse, REV) used in RT-PCR.

Gene	Primer Sequences (5′–3′)
*Hu.NOX4*	FOR: CCGGCTGCATCAGTCTTAAC
	REV: TCGGCACAGTACAGGCACAA
*Hu.Poldip2*	FOR:CAAAACAGAATGGAAAATATGAGACCGG
	REV: TGATTGATGCTCGTGACTGCCCA
*Hu.Col1A1*	FOR: ACTGGTGAGACCTGCGTGTA
	REV: AATCCATCGGTCATGCTCTC
*Hu.β-actin*	FOR: GTGCTATCCCTGTACGCCTC
	REV: GGCCATCTCTTGCTCGAAGT
*Mu.NOX4*	FOR: CATTCACCAAATGTTGGGC
	REV: TGCACACCTGAGAAAATACA
*Mu.Poldip2*	FOR: TCCTCAGAGGCTGGACATCT
	REV:ATGCTCAGAAGCCCACAGTT
*Mu.GAPDH*	FOR: CGTCCCGTAGACAAAATGGT
	REV: TTGATGGCAACAATCTCCAC

**Table 2 ijms-23-02999-t002:** Human and mouse sequences and accession numbers for miRNA primers used in RT-PCR.

Gene	Primer Sequences (5′–3′)
*has-miR-215*	ATGACCTATGAATTGACAGAC
*has-miR-9*	TCTTTGGTTATCTAGCTGTATGA
*has-miR-196b*	TAGGTAGTTTCCTGTTGTTGGG
*has-miR-203*	GTGAAATGTTTAGGACCACTAG
*has-miR-148a*	AAAGTTCTGAGACACTCCGACT
*has-U6*	CTCGCTTCGGCAGCACA
*has-miR-215*	ATGACCTATGATTTGACAGAC
*has-miR-9*	TCTTTGGTTATCTAGCTGTATG
*has-miR-196b*	TAGGTAGTTTCCTGTTGTTGGG
*has-miR-203*	GTGAAATGTTTAGGACCACTAG
*has-miR-148a*	AAAGTTCTGAGACACTCCGACT
*has-U6*	CTCGCTTCGGCAGCACA
*mmu-miR-215*	ATGACCTATGATTTGACAGAC
*mmu-miR-9*	TCTTTGGTTATCTAGCTGTATGA
*mmu-miR-196b*	TAGGTAGTTTCCTGTTGTTGGG
*mmu-miR-203*	GTGAAATGTTTAGGACCACTAG
*mmu-miR148a*	TCAGTGCACTACAGAACTTTGT
*mmu-U6*	GCTTCGGCAGCACATATACTAAAAT

## Data Availability

The data used to support this research are available from the corresponding author upon request.

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
