# Peer review of "MicroRNA 148a Suppresses Tuberculous Fibrosis by Targeting NOX4 and POLDIP2"

_ijms, 2022, doi:10.3390/ijms23062999_

Round 1
Reviewer 1 Report
In this paper, Woo et. al try to identify potential miRNA that regulates NOX4 in 56 tuberculous fibrosis.
Here are some comments:
Please explain that the number of samples (mice, patient samples; n=3, and n=6) is sufficient to draw conclusions. How were the sample sizes estimated? How man replicates was made?
In the introduction the last paragraph with the conclusion is unnecessary.
Increase the quality of figure 5.
Table 1 please center text.
Explain in statistic section and/or in figures description what it is *, **, #, ##.
Mistake in figure 4 description (double d, no e).
In figure 7 don’t understand why they are two lines from miRNA to NOX4/POLDIP2
Author Response
We would like to thank all of the editors and reviewers for helping us make a better revision. We revised our manuscript according to the comments and recommendations of the reviewers. We highlighted all changes in the revised manuscript in red letters.

Reviewer 2 Report
In the article under review the authors analyze the role of miRNA-148a in the regulation of NOX4 and POLDIP2 expression which involved in fibrosis into in vitro and in vivo (mice) models. The work was done at a high methodological level, but after reading a few questions/comments remained.
- The development of the protective immune response to BCG or M. tuberculosis is different, as well as very different immune responses to M. tuberculosis in humans and in a mouse model (especially for the TB-resistant C57BL/6 mouse strain). I recommend you mentioned that the data were obtained in a model system in all statements and conclusions in the article. It is not entirely correct to extrapolate the results to humans’ tuberculosis.
- You write: “After the NOX4 levels peaked, the expression of miR-148a peaked” (line 102). However, you don’t show this peak in Met5A in vitro models for either NOX4 or miR-148 (Fig. 3a-b). It is also not clear what is meant by a peak for miR-148 in an in vitro model of WT-mPMCs. Minimal expression? Could you please explain it?
- In the legend for Fig. 4, d is indicated for the second time instead of e.
- It would be more clearly show in the legend to the figure of the magnification. The bar shown in the figure merges with the picture.
- H&E staining visualizes only inflammation, but not fibrosis (see Materials and Methods section 4.10 – line 355-356). Fibrosis stain - van Gieson - which you mention in the legend to Figure 6.
- In the conclusion you write: “Clinically, pleural effusion miR-148a may potentially be a useful biomarker for the identification of tuberculous fibrosis” (line 213-214). In the present work, you didn’t investigate the miRNA expression profile of pleural effusion in TB patients. Are there any studies demonstrating the presence of this miRNA in TB patients? If so, please provide a link to such work. It is not correct to draw conclusions about clinical use based only on the model studies results.
Author Response

(The authors gave the same response as above.)

Round 2
Reviewer 1 Report
The recommended changes have been incorporated which have definitely improved the quality of this manuscript.